# Bioluminescent Optogenetics: A Novel Experimental Therapy to Promote Axon Regeneration after Peripheral Nerve Injury

**DOI:** 10.3390/ijms22137217

**Published:** 2021-07-05

**Authors:** Arthur W. English, Ken Berglund, Dario Carrasco, Katharina Goebel, Robert E. Gross, Robin Isaacson, Olivia C. Mistretta, Carly Wynans

**Affiliations:** 1Department of Cell Biology, Emory University School of Medicine, Atlanta, GA 30322, USA; dario.carrasco@emory.edu (D.C.); kg199911@gmail.com (K.G.); robin.isaacson@emory.edu (R.I.); ocmistr@emory.edu (O.C.M.); carly.wynans@emory.edu (C.W.); 2Department of Rehabilitation Medicine, Emory University School of Medicine, Atlanta, GA 30322, USA; 3Department of Neurosurgery, Emory University School of Medicine, Atlanta, GA 30322, USA; ken.berglund@emory.edu (K.B.); rgross@emory.edu (R.E.G.)

**Keywords:** axon regeneration, peripheral nerve injury, mice, electrophysiology

## Abstract

Functional recovery after peripheral nerve injury (PNI) is poor, mainly due to the slow and incomplete regeneration of injured axons. Experimental therapies that increase the excitability of the injured axons have proven remarkably successful in promoting regeneration, but their clinical applicability has been limited. Bioluminescent optogenetics (BL-OG) uses luminopsins, fusion proteins of light-generating luciferase and light-sensing ion channels that could be used to increase neuronal excitability if exposed to a suitable substrate. Excitatory luminopsins were expressed in motoneurons of transgenic mice and in wildtype mice transduced with adeno-associated viral vectors. Intraperitoneal administration of coelenterazine (CTZ), a known luciferase substrate, generated intense bioluminescence in peripheral axons. This bioluminescence increased motoneuron excitability. A single administration of CTZ immediately after sciatic nerve transection and repair markedly enhanced motor axon regeneration. Compound muscle action potentials were 3–4 times larger than controls by 4 weeks after injury. The results observed with transgenic mice were comparable to those of mice in which the luminopsin was expressed using viral vectors. Significantly more motoneurons had successfully reinnervated muscle targets four weeks after nerve injury in BL-OG treated mice than in controls. Bioluminescent optogenetics is a promising therapeutic approach to enhancing axon regeneration after PNI.

## 1. Introduction

Poor recovery from peripheral nerve injury (PNI) is a significant public health issue. As many as 20 million people in the United States are affected by nerve injuries resulting from trauma or medical disorders (NIH Publication No. 18-NS-4853). Individuals with PNI rarely regain complete nerve function, leading to a vast accumulation of persons with some degree of permanent disability [1]. Nerve injuries often lead to irreparable damage and decrease in quality of life [2]. 

Poor recovery from these injuries is largely due to the slow and inefficient process of axon regeneration and may be amplified by the fact that repair of the injured nerves, and subsequent target reinnervation, is often delayed by weeks or months [3]. The probability of successful functional reinnervation of peripheral nerves severely decreases the longer the repair is delayed [4,5]. Activity-dependent treatments, such as exercise or low frequency (20 Hz) electrical stimulation, are novel experimental therapies for PNI that have proven successful in preclinical studies [6,7], but they require specialized equipment and personnel that are not always easily accessible or affordable. In addition, surgical repair of nerve injuries is often delayed, as the traumatic events leading to these injuries may require immediate prioritization of other life-saving medical interventions. An approach to increasing the activity of injured neurons that might avoid these potential barriers inherent to clinical translation of activity-dependent therapies is needed. 

Bioluminescent optogenetics (BL-OG) could be one such novel approach. BL-OG utilizes luminopsin molecules expressed in neurons. These are fusion proteins of a light-generating luciferase and channelrhodopsin, a light sensitive cation channel (Figure 1). When exposed to a luciferase substrate, such as coelenterazine (CTZ), light is generated by the enzymatic action of the luciferase, producing direct neuronal excitation [8,9]. The goal of the work reported here was to evaluate the feasibility of BL-OG as a treatment for PNI. A preliminary report of some of these findings has been made [10].

## 2. Results

### 2.1. Point Mutation Renders Luminopsin Non-Functional without Compromising Expression Levels

As BL-OG involves multiple moving parts, including an exogenously expressed membrane protein (i.e., luminopsin), a luciferase substrate (i.e., CTZ), a solvent for CTZ, a CTZ metabolite (i.e., coelenteramide), and the resulting bioluminescence, it is critical to confirm that any observed effect is as intended and mediated by BL-OG, but not by any other byproducts. An ideal negative control would involve all these components yet would not transduce a membrane potential altering photocurrent. Using an in vitro heterologous expression system in human embryonic kidney (HEK) 293FT cells, we tested three mutations in the VChR1 moiety in the luminopsin molecule, analogous to *Chlamydomonas* channelrhodopsins known to render the channel non-functional by disrupting interaction with the chromophore, all-trans retinal [11,12]. All three mutations, namely E92R, R115A, and D248A, significantly reduced photocurrents to a saturating level of blue light compared to eLMO3 with wildtype VChR1 (Figure 2A). Quantification of peak current (Figure 2B) and steady-state current (Figure 2C) revealed that, among those three mutants, R115A was the most effective in terms of eliminating photocurrent (peak current: one-way analysis of variance (ANOVA) followed by Dunnett’s comparison to the control (eLMO3); *F*(3,20) = 12.5; * *p* = 0.0001; *n* = 6 cells each; *q*(20,4) = 4.85, 5.00, and 5.12; * *p* = 0.0003, 0.0002, and 0.0002 for E92R, R115A, D248A, respectively) (steady-state current: one-way ANOVA followed by Dunnett’s comparison to the control (eLMO3); *F*(3,20) = 11.3; * *p* = 0.0001; *n* = 6 cells each; *q*(20,4) = 4.62, 4.77, and 4.87; * *p* = 0.0005, 0.0003, and 0.0003 for E92R, R115A, D248A, respectively). We further confirmed that the reduction in photocurrent was not caused merely by a reduction in expression levels but rather a change in function, by measuring bioluminescence and fluorescence in a plate reader (Figure 2D,E, respectively). Expression levels when measured, either as bioluminescence or fluorescence, were overall comparable and were not significantly different (bioluminescence: *p* > 0.62; *F*(3,53) = 0.595; one-way ANOVA; *n* = 22, 11, 12, and 12 wells for eLMO3, E92R, R115A, and D248A, respectively)(fluorescence: *p* > 0.19; *F*(3,56) = 1.649; one-way ANOVA; *n* = 24, 12, 12, and 12 wells for eLMO3, E92R, R115A, and D248A, respectively). Accordingly, we chose the R115A mutant for subsequent viral vector production and used it as negative control for some in vivo experiments. 

### 2.2. Coelenterazine Produces Bioluminescence in Luminopsin Expressing Nerves

One of the advantages of BL-OG is, if somewhat obvious, the emission of bioluminescence. When used in vivo, it can confirm expression of luminopsin without sacrificing the animal and conducting postmortem histology. It can also be used to gauge dosage of CTZ and time course of the luminopsin action. To exploit this unique advantage of BL-OG, we subjected transgenic animals whose eLMO3 expression was limited to neurons and nerves expressing choline acetyl transferase (ChAT), including motoneurons, through a Cre-lox system, to whole-body bioluminescence imaging under anesthesia, following intraperitoneal injections of different doses of CTZ. Bioluminescence was observed throughout the body following systemic circulation of CTZ (Figure 3A). It was especially strong in the forebrain (consistent with widespread projection from the medial septum [13]), in the hindbrain (presumably in pedunculopontine nucleus, laterodorsal tegmental nucleus, and brainstem motor nuclei, bilaterally), in the fore and hind paws, in the stomach, in the kidneys, in the testes (in males), and in the tail. Bioluminescence was negligible in negative control male mice positive for only one of the two transgenes required for recombination and expression of eLMO3 (*n* = 2 animals). 

We observed various amplitudes and time courses of bioluminescence following injection of different doses of CTZ in dose-dependent manners (Figure 3B). Bioluminescence reached its peak within 5 to 40 min after injection and decayed gradually in the next hour or so. We observed a significant dose dependency in the amplitude of bioluminescence (Figure 3C; * *p* = 0.01; two-way ANOVA; *F*(2,12) = 6.75; *n* = 3 each for males and females). Although bioluminescence amplitude was slightly greater in males than in females, based on the results of our two-way ANOVA, we found no significant sex difference and also found that no significant interaction between sex and amplitude was present (*p* > 0.05; two-way ANOVA; *F*(1,12) = 4.74, *F*(2,12) = 0.27, respectively; n = 3 each for males and females). Bioluminescence with the highest dosage tested (20 mg/Kg) was significantly higher than the lower two dosages (5 and 10 mg/Kg; * *p* < 0.04; Tukey’s multi comparisons; *q*(3,15) > 3.87; *n* = 6 mice (3 males and 3 females)).

### 2.3. BL-OG Treatments Increase Motoneuron Excitability

Spinally evoked muscle potentials (SEMPs) were elicited by electrical stimulation applied at the first lumbar vertebra. As observed in rats [14] and in human subjects [15,16], these SEMPs were noted as three components that have been termed early response (ER), middle response (MR), and late response (LR) (Figure 4A). These responses are thought to be the result of activation of different spinal pathways. The ER has been proposed to be the result of direct motoneuron stimulation and the MR to monosynaptic effects. The LR is described as due to activation of polysynaptic pathways exciting the motoneurons and occurs only at higher stimulus intensities than the ER and MR [14]. In the present study, the amplitude of both the ER and MR increased, with increases in the intensity of the applied stimulation in a manner consistent with their proposed activation schemes. The ER recruitment curve (Figure 4B: black symbols) was very much like a typical sigmoidal M response recruitment curve and that of the MR (Figure 4B: open symbols) was very much like the gradually increasing and stabilizing monosynaptic H reflex recruitment curve recorded in mice [17].

In six tamoxifen treated SLICK-A mice (three males, three females), eLMO3 expression in neurons innervating the GAST muscles was induced by injecting an AAV2/9 vector encoding a Cre dependent luminopsin into one sciatic nerve. The contralateral nerve was not injected. Four weeks later, SEMPs were evoked in both GAST muscles and stimulus-response curves were generated as described above. Following CTZ injection, no significant change in background EMG activity was noted over the time-course of anticipated bioluminescence studied (Figure 4C), but a gradual leftward shift in the stimulus response curve for the ER was observed in muscles on the side of the mouse that had been induced to express eLMO3, indicating that ERs of similar amplitude were evoked at lower stimulus intensity after CTZ treatment. This shift peaked at around 45 min after CTZ injection and by 3 h after injection the ER stimulus response curve had returned to that seen prior to treatment (Figure 4D), consistent with the time course of bioluminescence emission in vivo (Figure 3B). The stimulus intensity needed to evoke an ER amplitude of 40% of maximum, a clinically used metric of axonal excitability [18], decreased by 16.32% ± 3.85% (Mean ± SEM) with CTZ treatment (Figure 4E). No significant sex difference was observed (unpaired test, t_4_ = 0.368, not significant). No significant change was found after CTZ treatment in the ER amplitude of the contralateral muscle, whose motoneurons had not been induced to express eLMO3 (Figure 4F). The BL-OG treatment thus increased motoneuron excitability.

### 2.4. Effects of BL-OG Treatment on Axon Regeneration in Transgenic Mice

Male and female mice of three genotypes (ChAT-Cre, eLMO3, and ChAT-Cre/eLMO3) were treated a single time with either CTZ or the CTZ solvent (Fuel-Inject, referred to hereafter as “Fuel”) immediately after repair of the cut sciatic nerve (Figure 5A). The extent of successful motor axon regeneration and muscle reinnervation was studied in these six groups four weeks later using maximal direct muscle responses (Mmax) in the GAST and TA muscles to stimulation of the sciatic nerve proximal to the injury. Examples of Mmax responses recorded from reinnervated GAST muscles in two different ChAT-Cre/eLMO3 mice, four weeks following transection and repair of the sciatic nerve, are shown in Figure 5B. One mouse was treated at the time of injury with CTZ (top) and the other with Fuel (bottom). The larger amplitude and shorter latency responses seen in the CTZ treated mice are consistent with earlier and more mature muscle reinnervation. 

We evaluated whether male and female mice responded similarly to BL-OG treatment by comparing Mmax amplitudes in the two muscles between males and females in the same groups using a two-way (sex and genotype/treatment) ANOVA. For both muscles there was a significant effect of genotype/treatment (*p* < 0.01, *F*(5, 30) = 4.85 for GAST; *p* < 0.01, *F*(5, 30) = 9.28 for TA), but no significant effect for sex *p* = 0.49, (*F*(4, 30) = 0.87 for GAST; *F*(4, 30) = 0.14, *p* = 0.97 for TA), and no significant effect for interaction (*p* = 0.84, *F*(20, 30) = 0.65 for GAST; *p* = 0.24, *F*(20, 30) = 1.33 for TA). Based on these results, we combined data from the two sexes for all subsequent analyses. Data from these experiments are summarized in Figure 5C. The Mmax recorded from each muscle in each tested animal four weeks after sciatic nerve transection and repair was scaled to the mean of untreated wildtype mice studied at the same four week survival period [19], which is represented by the horizontal dashed line present at 1.0. A one-way ANOVA was conducted on the combined data for each muscle and the results were significant, both for GAST (*p* < 0.001, *F*(5, 56) = 5.22) and TA (*p* < 0.0001, *F*(5, 59) = 8.77). Using post hoc paired testing (Fisher’s LSD), significant differences (*p* < 0.05) were found for both muscles between the ChAT-Cre/eLMO3 mice that had been treated with CTZ and all the other treatment groups. Thus, a single treatment with the luciferase substrate, CTZ, resulted in enhanced motor axon regeneration and muscle reinnervation only in mice expressing the excitatory luminopsin. 

### 2.5. Effects of BL-OG Treatment on Axon Regeneration after Viral eLMO3 Transduction

Ten sciatic nerves from SLICK-A mice (six male, four female) were injected with a viral vector encoding a Cre-dependent eLMO3 construct (Figure 6A). Recombination through inducible Cre recombinase was then induced by tamoxifen treatment to express eLMO3 in a subset of motoneurons. Four weeks after transection and repair of the sciatic nerve, and treatment with either CTZ (10 mg/Kg, i.p.) or Fuel, M responses were recorded from the reinnervated GAST and TA muscles. Amplitudes of Mmax were scaled to those recorded from untreated animals [19]. Results of comparison of CTZ-treated and Fuel-treated animals are shown in Figure 6B. A one-way ANOVA applied to these data returned a significant result (*p* < 0.003, *F*(3, 16) = 7.172). In post hoc paired testing, significant differences between the two treatment groups were found for both muscles (Figure 6B). BL-OG treatment in the motoneurons induced to express eLMO3 resulted in enhanced motor axon regeneration and muscle reinnervation.

In an additional group of six mice, one sciatic nerve was injected with an AAV2/9 vector encoding eLMO3 under control of the *Hsyn* promoter, and the contralateral nerve was similarly injected but with a vector encoding a mutant eLMO3, in which the channelrhodopsin portion of the fusion protein was made unresponsive to light by a single amino acid replacement (R115A, see Figure 2). Four weeks after sciatic nerve transection and repair and a single treatment with CTZ, M responses were recorded from the reinnervated GAST and TA muscles, bilaterally. The amplitudes of Mmax were scaled, as above. Results are shown in Figure 6C. In both GAST and TA muscles, whose motoneurons had been induced to express the wildtype eLMO3, scaled Mmax amplitudes four weeks later were more than three times larger than those found in untreated animals and significantly greater than the scaled Mmax amplitudes recorded from contralateral muscles, whose motoneurons were induced to express the mutated eLMO3 (ANOVA, *p* = 0.04, *F*(3, 20) = 3.407, post hoc GAST *p* < 0.04, TA *p* < 0.03). The scaled responses from the muscles on the side of the mice induced to express the mutated luminopsin were also not significantly different from unity, the mean M response amplitude found in untreated mice, indicating that the CTZ treatments had been ineffective in promoting motor axon regeneration in these controls. 

### 2.6. Retrograde Labeling

Four weeks after bilateral sciatic nerve transection, and repair in SLICK-A mice that had been induced to express a Cre-dependent luminopsin in motoneurons using AAV2/9 injection into only one sciatic nerve, motoneurons that had regenerated their axons successfully and reinnervated the GAST muscles were labeled with a fluorescent retrograde tracer. In these mice, Cre recombination had been induced in motoneurons also expressing YFP by prior tamoxifen treatment. This experimental approach resulted in retrograde labeling of two populations of motoneurons. Some motoneurons were marked by the red fluorescent tracer only. We assumed that axons of these motoneurons had regenerated successfully, but they did not express Cre recombinase and, therefore, could not express eLMO3. Other motoneurons were marked both by the retrograde tracer and YFP; they were double labeled. Examples of these two kinds of motoneurons are shown in Figure 7A.

The mean (±SEM) numbers of motoneurons in these two categories are shown in Figure 7B. The number that contained only the retrograde tracer was not significantly different (t(3) = 3.101, not significant) on the two sides of these mice, but the number of motoneurons containing both the retrograde label and YFP on the sides of the mice in which eLMO3 expression had been induced, was twice that found on the sides of the mice that did not express the luminopsin (t(3) = 10.60, *p* < 0.002). The significantly greater (t(3) = 7.248, *p* < 0.01) total number of labeled neurons (single + double labeled) found there can be attributed to the BL-OG treatment. 

Prior to euthanizing these mice, M responses were recorded from the GAST and TA muscles on the two sides of the animals. Scaled Mmax values from these experiments are shown in Figure 7C. For both TA and GAST, scaled Mmax values were significantly greater (*p* < 0.01, *F*(3, 11) = 6.449, post hoc GAST *p* < 0.01, TA *p* < 0.05) on the sides of the mice in which eLMO3 expression had been induced. These results are consistent with the results of the retrograde labeling. BL-OG treatment increased the number of motoneurons whose axons had regenerated successfully.

## 3. Discussion

The development of experimental therapies that enhance axon regeneration could advance the treatment of PNI, injuries which are common in the United States today. The functional recovery of patients suffering from PNI remains poor, largely because of the slow and inefficient nature of axon regeneration [3], but also because this process slows precipitously at longer times after injury [4,5]. The activation of injured sensory and motor neurons has proven especially effective in enhancing axon regeneration in pre-clinical animal models [6,7]. However, the ways of increasing the activity of the injured neurons in the variety of injuries encountered clinically has posed barriers to translating these findings. The ability to generate direct neuronal excitation using BL-OG could provide a basis for scaling these barriers. The goal of this study was to investigate the feasibility of using BL-OG in this context. 

The results presented here show that CTZ treatments of mice expressing an excitatory luminopsin will generate bioluminescence and increase the excitability of motoneurons over a similar time course. We also show that this increase in excitability is both sufficient to promote motor axon regeneration after PNI, but that it also requires a fully functional luminopsin for the enhancement. The magnitude of evoked M responses in the reinnervated TA and GAST muscles recorded four weeks after PNI and a single CTZ treatment was significantly larger than found in controls, whether the luminopsin was expressed in transgenic mice or induced by a viral vector. We interpret this finding to mean that the BL-OG treatment stimulated the elongation of regenerating motor axons and their reinnervation of muscle targets. No significant effect was found with the same treatment if the luminopsin expressed had been mutated, such that it could not increase neuronal excitability with CTZ treatment. In addition, BL-OG treatments significantly increased the number of motoneurons, whose axons regenerated successfully, an equally important metric of enhancing axon regeneration after PNI. 

We also show that these effects of BL-OG are the same in both males and females. This outcome is in marked contrast to the significant sex differences we have found in the requirements for exercise to enhance axon regeneration successfully [20]. Thus, we project similar treatment approaches for males and females in any future clinical application of BL-OG as a treatment for enhancing functional recovery following PNI. 

The results of the experiments described above support the idea that BL-OG is an effective method to promote motor axon regeneration and muscle reinnervation through its activation of the injured neurons. More precisely, these results are consistent with an interpretation that it is the increase in excitability of the motoneurons that produces this enhancement of regeneration. Treatment with CTZ reduced the threshold for the early response to epidural stimulation that is widely regarded as a measure of direct motoneuron activation [14]. Because the treatments did not increase the amplitude of background EMG activity over the period of induced bioluminescence, it is reasonable to assume that this bioluminescence resulted in a subthreshold increase in motoneuron membrane potential and that was adequate to stimulate axon regeneration. We have argued elsewhere [21,22,23] that such modest increases in excitability may be all that is required to place or maintain axotomized motoneurons into a pro-regenerative state. Achieving this modest level of motoneuron activation may be especially important when applied immediately after nerve repair, before injury-induced increases in neuronal excitability and regeneration state may be more widespread [24].

The results described above do provide fundamental evidence that BL-OG could be an effective treatment for PNI. However, they are limited to experiments in which neurons were induced to express eLMO3 before an injury, an impossible clinical scenario. To expand upon these results and move this experimental therapy closer to actual translation, it will be important to study the effectiveness of BL-OG when luminopsin expression is induced after the injury. We believe that our experiments using virally mediated gene delivery provide an effective model for inducing luminopsin expression in injured motoneurons, but future studies will need to extend these models to post-injury viral application. These studies also must include consideration of sensory and post-ganglionic sympathetic axon regeneration, as well as motor recovery. The experiments described here also used a simple repair of a transected nerve, so that regenerating axons need only traverse the injury site before entering a regeneration pathway in the distal segment of the cut nerve. Future studies also might consider the effectiveness of BL-OG in promoting axon regeneration across gaps, a more challenging clinical scenario. Because the mechanism of action of BL-OG involves neuronal activity-dependent promotion of axon outgrowth, BL-OG stimulation of axon growth across a gap seems feasible. Treatments of PNIs with BL-OG are feasible and these future studies have the potential to move it toward addressing a major public health issue. 

## 4. Materials and Methods

### 4.1. Molecular Biology

The luminopsin construct (Figure 1) is from *Volvox* channelrhodopsin 1 (VChR1), a red-shifted, highly sensitive channelrhodopsin variant, fused with *Gaussia* luciferase (GLuc) through a short linker [25], and slightly modified to include the trafficking signal from a neuronal potassium channel for better membrane targeting in front of the enhanced yellow fluorescent protein (EYFP) tag (enhanced LMO3 or eLMO3) [12]. As a control, we introduced 3 non-functional point mutations in the VChR1 moiety in eLMO3 using site-directed mutagenesis, similar to a previous publication [26]. They were glutamate 92-to-arginine (E92R, analogous to E97R in *Chlamydomonas* ChR2 [11]), arginine 115-to-alanine (R115A), and aspartate 248-to-alanine (D248A) (analogous to R119A and D292A in *Chlamydomonas* ChR1/ChR2 chimera, respectively) [12]. The 3 mutant eLMO3s were subcloned into a modified pcDNA3.1 plasmid with the CAG promoter for general expression in mammalian cells in vitro using standard molecular biology methods, including restriction enzymes and ligase. One select mutant, namely R115A, was further incorporated into an adeno-associated virus (AAV) 2 transfer plasmid with the human synapsin 1 promoter (Hsyn) for viral vector production for in vivo use. 

### 4.2. Tissue Culture and Virus Production

The efficacy of point mutations was assessed using a heterologous expression system through patch-clamp electrophysiology of photocurrents and plate-reader assays of bioluminescence and fluorescence. Human embryonic kidney (HEK) 293FT cells were maintained in Dulbecco’s modified Eagle’s medium (D6171SigmaChemical Co., St. Louis, MO, USA), supplemented with fetal bovine serum (10% *v*/*v*; S1195,Atlanta Biologicals, Atlanta, GA, USA), l-glutamine (2 mM final; G7513, Sigma), and the antibiotic G-418 sulfate (500 µg/mL final; BioWorld 40710026, Irving, TX, USA) in a humidified incubator at 37 °C with 5% CO_2_. A day before transfection, HEK cells were seeded onto 15 mm diameter coverslips in a 12 well plate and a 96 well black plate with clear bottoms (GBO 655090, Thermo Fisher, Waltham, MA, USA) for electrophysiology and plate reader assays, respectively. Lipofection was conducted following the manufacturer’s instructions (Lipofectamine 3000, Invitrogen, Thermo Fisher, Waltham, MA, USA).

HEK293FT cells were also used for virus production. A Cre recombinase-dependent eLMO3 construct under control of the ubiquitous elongation factor 1α (EF1α) promoter (Addgene plasmid #: 114105; a generous gift from Dr. Ute Hochgeschwender) was packaged into an adeno associated viral (AAV) vector pseudotyped with AAV9 capsid protein. AAV vectors were produced in Emory Viral Vector Core or in-house as recombinant proteins using the standard calcium phosphate transfection of 3 plasmids followed by purification through ultracentrifugation with an iodixanol gradient. Titer was determined to be >10^13^ viral genomes/mL using quantitative PCR.

### 4.3. Electrophysiology

Patch-clamp recordings were conducted under an upright wide-field fluorescence microscope (Scientifica HyperScope) equipped with an LED (CoolLED pE-300ultra, CoolLED, Andover, MA, USA), a multi-bandpass filter cube (Chroma 89402, Chroma Technology, Bellows Falls, VT, USA), water immersion objective (Nikon CFI75 LWD 16X W 0.8NA), a scientific CMOS camera (OptiMOS, QImaging, Surrey, BC, Canada), a micromanipulator (Scientifica PatchStar, Scientifica, Clarksburg, NJ, USA), a patch-clamp amplifier (Axon MultiClamp 700B, Molecular Devices, San Jose, CA, USA), and a digitizer (Axon Digidata 1550B, Molecular Devices) at room temperature. eLMO3- and its mutant-expressing HEK cells were identified through the EYFP tag under blue excitation (460 nm). The same light was used to activate photocurrent. Whole-cell voltage-clamp was obtained at −60 mV using a pipette pulled from borosilicate glass capillaries (Sutter Instruments, Novato, CA, USA) containing 140 mM K-gluconate, 2 mM MgCl_2_, 0.5 mM CaCl_2_, 4 mM Na_2_-ATP, 0.4 mM Na_3_-GTP, 5 mM EGTA, and 10 mM HEPES, and pH was adjusted to 7.15 with KOH. The external solution contained 150 mM NaCl, 3 mM KCl, 2 mM MgCl_2_, 2 mM CaCl_2_, 10 mM HEPES, and 20 mM glucose, and pH was adjusted to 7.35 with NaOH. The pipette had a resistance between 5 and 7 MΩ when filled with the internal solution. Capacitive currents and series resistance were compensated by 75–85%. Data were digitally filtered at 3 kHz and digitized at a sampling rate of 10 kHz. A patch-clamp software was used to control the recordings and the photostimulus (Axon pClamp 10, Molecular DevicesData were processed and analyzed using in-house and NeuroMatic [27] macros in Igor Pro 8 (WaveMetrics, Portland, OR, USA). 

### 4.4. Plate-reader Assays

Bioluminescence and fluorescence were quantified using a plate reader with environmental control (Cytation 5, BioTek, Winooski, VT, USA). Native coelenterazine (CTZ; NanoLight Technologies 303, Pinetop, AZ, USA) was first dissolved into a proprietary solvent (NanoLight Technologies NanoFuel, Pinetop, AZ, USA) at 50 mM, diluted into the culture media, and added to each well (50 µM final) immediately before the assay. 

### 4.5. Animals and Luminopsin Expression

All procedures in this study were approved by the Institutional Animal Care and Use Committee of Emory University, and conformed to the Guidelines of the Office of Laboratory Animal Welfare of the National Institutes of Health. All mice used in experiments were on a C57BL/6 background and were genotyped by Transnetyx, Inc. prior to inclusion. 

Two methods were used to generate expression of the luminopsin construct: selective breeding of a transgenic mouse line and adeno-associated viral vector-mediated gene delivery. In the transgenic mice used in this study, the transgene encoding eLMO3 under control of the ubiquitous CAG promoter, was placed immediately downstream from a STOP sequence flanked by two loxP sites (floxed). The entire cassette was inserted to the Rosa26 locus through homologous recombination. These transgenic mice were obtained as a generous gift from Dr. Ute Hochgeschwender. The development of these mice is described elsewhere [28]. In the experiments reported here, the mice were bred with mice expressing Cre recombinase under control of the promoter for choline acetyl transferase (ChAT-Cre) (JAX 006410). In the spinal cords of these mice, the Cre recombinase will be restricted to motoneurons and a few interneurons [29]. Offspring from this breeding were genotyped using respective JAX protocols, and animals that were positive for both the eLMO3 construct and the ChAT-Cre genes (ChAT-Cre/eLMO3) were selected for experiments. Offspring that expressed Cre recombinase but not the floxed eLMO3 allele (ChAT-Cre) or expressed the floxed allele but no Cre recombinase (eLMO3) were used as controls. A total of 60 transgenic mice (30 male and 30 female) were studied in axon regeneration experiments. ChAT-Cre/eLMO3 mice were also used in in vivo bioluminescence experiments (*N* = 3 each for males and females).

Because we wanted to begin to consider whether BL-OG could someday be used clinically, we also expressed eLMO3 in injured neurons using two slightly different variations of the same viral vector. In one of these viral vectors, a Cre-dependent eLMO3 construct under control of the ubiquitous EF1α promoter was packaged into an AAV2/9 vector. To induce expression in injured neurons, this vector was injected into the sciatic nerves of mice known as Single-neuron Labeling with Inducible Cre-mediated Knockout or SLICK. These mice express a tamoxifen-inducible Cre recombinase and enhanced Yellow Fluorescent Protein (eYFP) under control of the Thy-1 promoter [30]. In particular, we used the A strain of SLICK mice (SLICK-A), in which the transgene is expressed only in a subset of motoneurons and sensory (dorsal root ganglion, DRG) neurons. 

In isoflurane anesthetized 6–8-week-old SLICK-A mice, the sciatic nerves were exposed in the mid-thigh and injected with one microliter (10^9^ viral genomes) of an AAV2/9 vector carrying the Cre-dependent eLMO3 construct using a Hamilton syringe connected to a 35G needle. The injection was performed slowly, and the needle left in place for 15 min after the completion of the injection. Four weeks later, to allow for retrograde transport of the virus and neuronal transduction, the injected mice were treated with tamoxifen (Sigma-Aldrich, St. Louis, MO, USA) to induce Cre recombinase expression. The Cre recombinase enzyme was activated by 2 distinct oral tamoxifen treatment regimens, each consisting of 3 daily treatments. The treatments were separated by 2 weeks. The efficacy of this tamoxifen treatment protocol to induce Cre recombinase translocation in SLICK-A mice has been validated previously [31]. 

In a second line of experiments using viral-mediated gene delivery to induce eLMO3 expression, 1 sciatic nerve in each of 6 wildtype mice (3 male, 3 female) was injected with an AAV2/9 vector encoding eLMO3 under control of the neuronal Hsyn promoter, to restrict expression to neurons. The contralateral sciatic nerves of these mice were injected with vector encoding the mutant eLMO3, R115A, also under control of *Hsyn*. 

### 4.6. Bioluminescence Experiments

Pharmacodynamics of CTZ was assessed through in vivo bioluminescence imaging in the double transgenic mice. Under isoflurane anesthesia (2% in oxygen), shaved mice received an intraperitoneal injection of a different dose of CTZ (Inject-A-Lume) using an insulin syringe. Immediately after injection, they were placed into an imaging dark box (Fujifilm LAS-3000, Tokyo, Japan) equipped with a large aperture (f/0.85) lens with no filter, a cooled CCD camera, a heating pad, and isoflurane and vacuum lines. Ultrabinning was used with consecutive 1 min exposures for up to an hour. Imaging sessions were repeated in the same animals but with different dosages of CTZ. Data were processed and analyzed using in-house macros in Igor Pro. In each animal, a region of interest was drawn over its entire body and background subtracted bioluminescence signal was calculated for each frame. 

### 4.7. Motoneuron Excitability Experiments

Six SLICK-A mice (three males, three females) were induced to express a Cre-dependent eLMO3 by unilateral injection of a viral vector into the sciatic nerve (see above). Two weeks after the last tamoxifen treatment to induce Cre recombinase expression, the effect of CTZ treatment on motoneuron excitability was studied. In isoflurane anesthetized animals, bipolar fine wire EMG electrodes [32] were inserted into the left and right lateral gastrocnemius (GAST) muscles. Two needle electrodes were constructed from 5 mm long insect pins (Minuten, Carolna Biological Item # 654366, Burlington, NC, USA) micro welded to teflon-coated stranded stainless steel wire (AS631, Cooner Sales, Chatsworth, CA, USA). The welded areas were insulated with epoxy, but the rest of the pin was not. The electrodes were placed on either side of the spinous process of the first lumbar vertebra and lowered until their tips were in contact with the lamina of that vertebra. Background EMG activity was monitored continuously from the two GAST muscles and, when it was maintained at a consistent baseline level for 20 ms, a short (0.3 ms) constant voltage stimulus was delivered through the electrodes using custom Labview software. The EMG activity in the two muscles, sampled at 10 KHz, was recorded from 20 ms prior to the stimulus until 100 ms after delivery, and recorded to disc. Stimuli were delivered no more often than once every 3 seconds to avoid fatiguing the muscles. As noted with similar stimulation in rats [14], three components of the spinally evoked muscle potentials (SEMPs) were noted (Figure 4A). An early response (ER) was found with a latency of 1–2 ms, as were a middle response (MR, 3–4 ms latency) and a late response (LR, >6 ms latency. Stimulus intensity was then increased gradually until the amplitude of these potentials reached a maximum. The mouse then received an injection of CTZ (Inject-A-Lume, NanoLight Technologies, Inc., 10 mg/Kg, i.p.) and the procedure repeated. Full stimulus response curves were generated every 15 min for 4 hours. At the end of the experiment, mice were euthanized with Euthasol solution.

### 4.8. Nerve Transection Surgery

The sciatic nerve was cut and surgically repaired in 80 mice. In isoflurane anesthetized animals, the sciatic nerve was exposed in the mid-thigh and placed on a small sheet of medical grade silastic (Dow Corning, catalog number: 501–1, VWR, Radnor, Pennsylvania). Application of 2 µL of a fibrin glue mixture, made from fibrinogen, fibrin, and thrombin [33], secured the nerve to the silastic sheet. Once the glue cured, forming a clot, the sciatic nerve was carefully transected and the freshly cut ends were aligned before another 2 µL of glue was applied [34]. The incision was then sutured closed, and antibacterial cream was applied to the skin. Immediately after wound closure, the mouse was treated, either with CTZ (10 mg/Kg, i.p.) or with a similar volume of a CTZ solvent (Fuel). 

### 4.9. M Response Recovery

In all the transgenic mice studied, and in five (3 male, 2 female) tamoxifen-treated SLICK-A mice induced to express Cre-dependent eLMO3 by *bilateral* virus injection into the sciatic nerves, both sciatic nerves were cut and repaired, as described above, and motor axon regeneration was studied using evoked electromyographic (EMG) activity. Four weeks after nerve transection and repair, the sciatic nerves were exposed bilaterally in isoflurane anesthetized animals. Two monopolar needle electrodes (Ambu #74325-36/40, Columbia, MD, USA) were placed next to each nerve as it exited the pelvis. Bipolar fine wire EMG electrodes were inserted into the lateral gastrocnemius (GAST) and tibialis anterior (TA) muscles of both hindlimbs. Ongoing EMG activity recorded from these electrodes was sampled at 10 KHz using a custom Labview program, and when it was maintained at a consistent baseline level for 20 ms, a short (0.3 ms) constant voltage pulse was delivered to the stimulating electrodes. The evoked EMG activity was recorded from 20 ms prior to the stimulus until 100 ms after the stimulus and saved to disc. A compound muscle action potential or direct muscle (M) response was recognized as a triphasic potential at short latency. Stimulus intensity was then increased gradually until a maximum M response (Mmax) was elicited in both muscles. Stimuli were delivered no more often than once every 3 seconds to avoid fatiguing the muscles. The mean rectified amplitude of Mmax in the reinnervated muscles was scaled to that recorded in the same manner from untreated mice 4 weeks after sciatic nerve transection and repair [19].

### 4.10. Retrograde Labeling

In four (2 male, 2 female) tamoxifen-treated SLICK-A mice, in which only 1 sciatic nerve had previously been injected with an AAV2/9 vector expressing a Cre-dependent eLMO3, the sciatic nerve was cut and repaired (see above) bilaterally. Animals were treated at that time with a single injection of CTZ (10 mg/Kg, i.p.). Four weeks later, retrograde tracers were injected into both GAST muscles to mark the motoneurons whose axons had regenerated and successfully reinnervated the muscle. Two microliters of a 1% solution of cholera toxin B conjugated to Alexa Fluor 555 (Invitrogen, Eugene, OR) was injected into each GAST muscle using a Hamilton syringe fitted with a 35G needle. Injections were made in several places in the muscles and left in place for 5 minutes between injections. Three days later, the mice were euthanized and perfused with a 4% paraformaldehyde solution. Lumbar spinal cords were harvested, cryoprotected in sucrose solution, sectioned in a transverse plane on a cryostat at 40 µm thickness, mounted onto charged glass slides, and cover slipped using Vectashield (Vector Laboratories, Burlingame, CA, USA). Images of the ventral horns of each section were captured using a Leica DM6000 upright microscope, a Hamamatsu low-light camera, and HCImage software. Neurons were then manually scored as retrogradely labeled if the Alexa Fluor 555 (red) label filled the neuron soma and extended into its proximal dendrites and contained a visible nuclear area devoid of label. To avoid double counting, care was taken to identify any fragments of each labeled neuron in adjacent serial sections. Retrogradely labeled neurons were also scored as containing YFP and, thus, expressing Cre recombinase or not. Comparison of counts of singly (Alexa Fluor 555 only) and double labeled motoneurons were compared between the two sides of the animals using a paired t-test.

Prior to euthanasia and tissue harvesting, M responses were recorded from both GAST muscles in these mice. Comparison of scaled Mmax amplitudes on the two sides of the mice was performed using ANOVA, with post hoc paired testing.

## Figures and Tables

**Figure 1 ijms-22-07217-f001:**
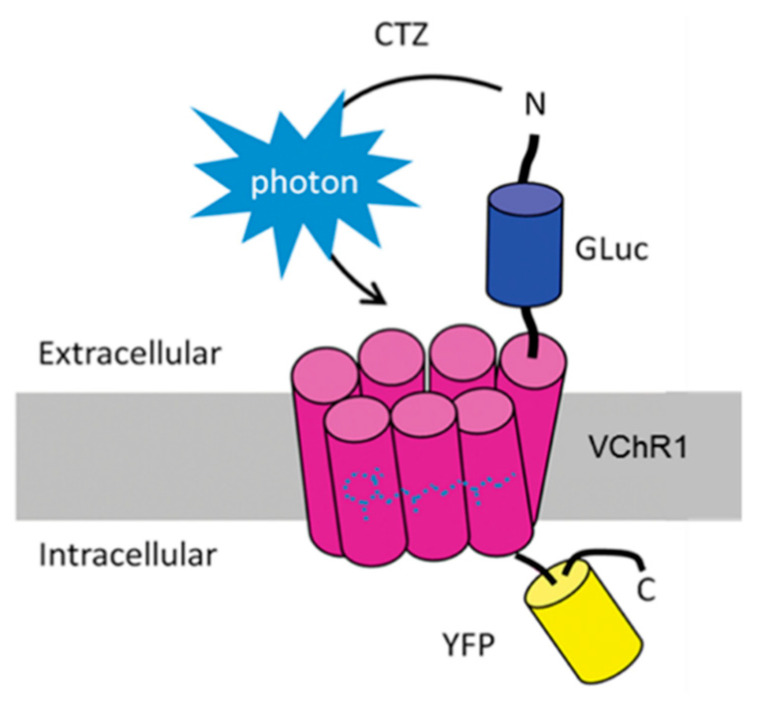
Diagram of the excitatory luminopsin, enhanced luminopsin 3 (eLMO3), used in bioluminescent optogenetics (BL-OG) experiments. Luciferase enzyme from the marine copepod, *Gaussia princeps* (GLuc) was fused to a red shifted channelrhodopsin derived from *Volvox carteri* (VChR1) and coupled to an intracellular yellow fluorescent protein (YFP) at its cytosolic tail. Bioluminescence is evoked by treatments with the luciferase substrate, coelenterazine (CTZ), opening the light-sensitive cation channel of VChR1.

**Figure 2 ijms-22-07217-f002:**
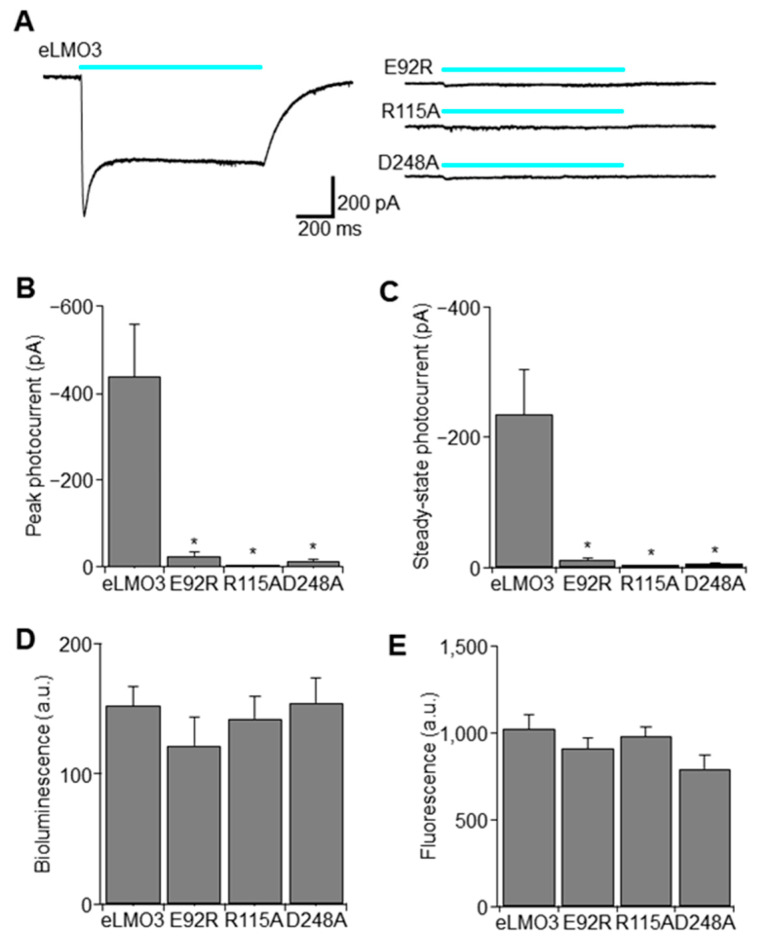
Mutant luminopsins with negligible photocurrent without altering expression levels. (**A**) Representative photocurrent traces to photostimulation (cyan bars) obtained from (HEK) 293FT cells expressing a different luminopsin indicated. The scale bars apply to all the traces. (**B**) Peak photocurrent. Asterisks (*) indicate significant differences from the wildtype eLMO3 (see text). Error bars indicate SEM in this and subsequent figures. *N* = 6 cells each. (**C**) Steady-state current measured in the period where photocurrent was inactivated during photostimulation. Asterisks indicate significant differences from the wildtype eLMO3 (see text). (**D**) Bioluminescence measured with 50 µM CTZ using a plate reader. *N* = 22, 11, 12, and 12 wells, respectively. (**E**) Fluorescence measured using a YFP filter cube. *N* = 24, 12, 12, and 12 wells, respectively.

**Figure 3 ijms-22-07217-f003:**
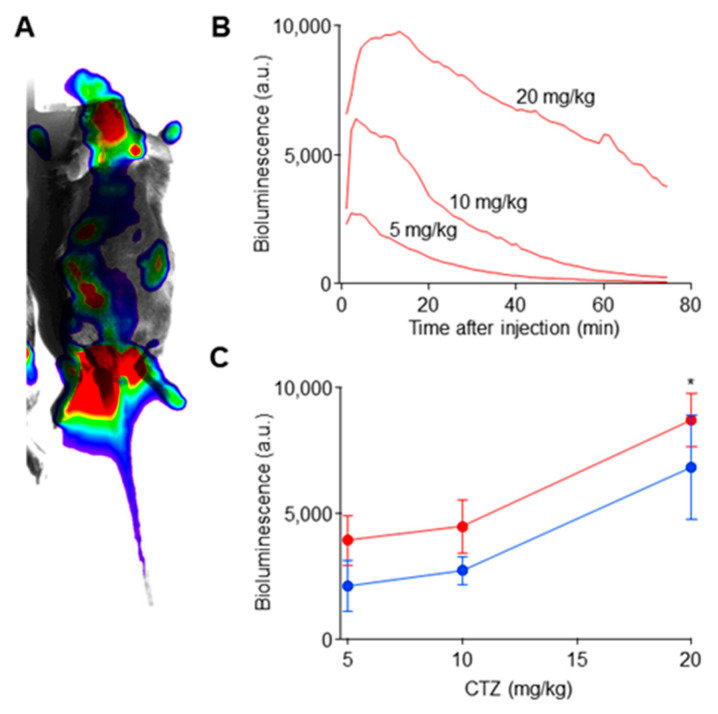
In vivo bioluminescence imaging in ChAT-Cre/eLMO3 transgenic mice following intraperitoneal injection of CTZ. (**A**) A representative image of bioluminescence (in pseudo color) superimposed onto a bright-field image (gray). A background subtracted bioluminescence image was averaged between 5 and 40 min after CTZ injection and thresholded to show only significant signal. (**B**) Representative time courses of bioluminescence following injections of different dosage of CTZ. The same female was subjected to the three different doses. (**C**) Quantification of peak bioluminescence. An asterisk indicates significant difference from the rest (see text). *N* = 3 males (in blue) and 3 females (in red).

**Figure 4 ijms-22-07217-f004:**
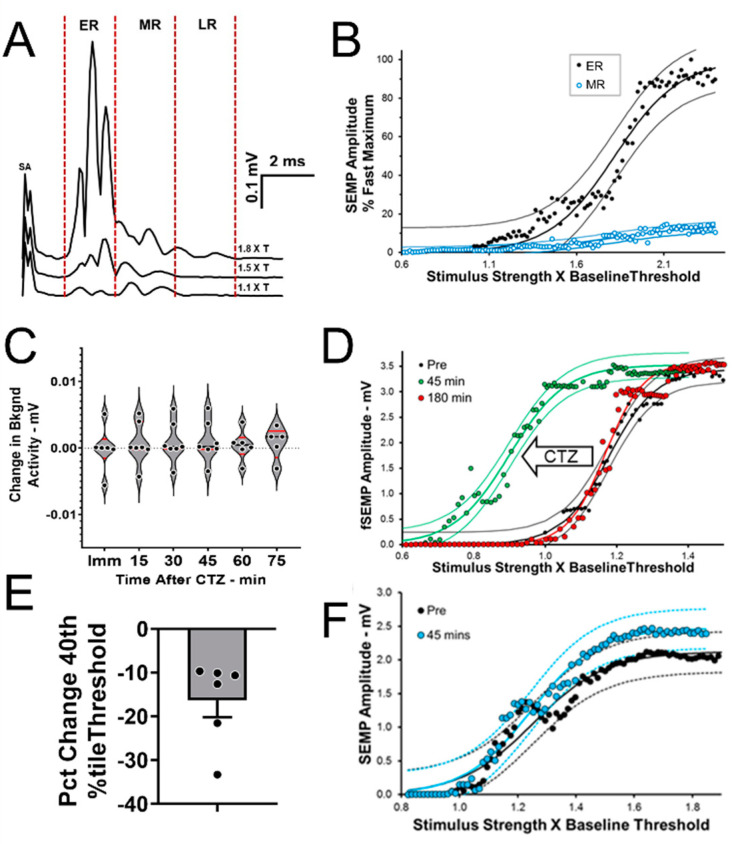
BL-OG increases motoneuron excitability. (**A**) Spinally evoked motor potentials (SEMPs) evoked from epidural stimulation at the level of the first lumbar vertebra in a mouse induced to express eLMO3 in sciatic motoneurons contain early (ER), middle (MR), and late (LR) responses. (**B**) Amplitudes of both ER and MR increase with increasing stimulus intensity, albeit differently. Responses were fit with a sigmoidal curve (+95% confidence intervals). (**C**) Administration of coelenterazine (CTZ, 10 mg/Kg, i.p.) did not change ongoing EMG activity significantly. (**D**) CTZ produced a leftward shift in the ER stimulus-response curve in the gastrocnemius muscle on the side of the mouse expressing eLMO3. (**E**) In six mice, this dose of CTZ reduced the threshold for ER activation by 16.32% (+3.85%, SEM). (**F**) In similar data from the contralateral gastrocnemius muscle, CTZ treatment had no effect on motoneuron excitability.

**Figure 5 ijms-22-07217-f005:**
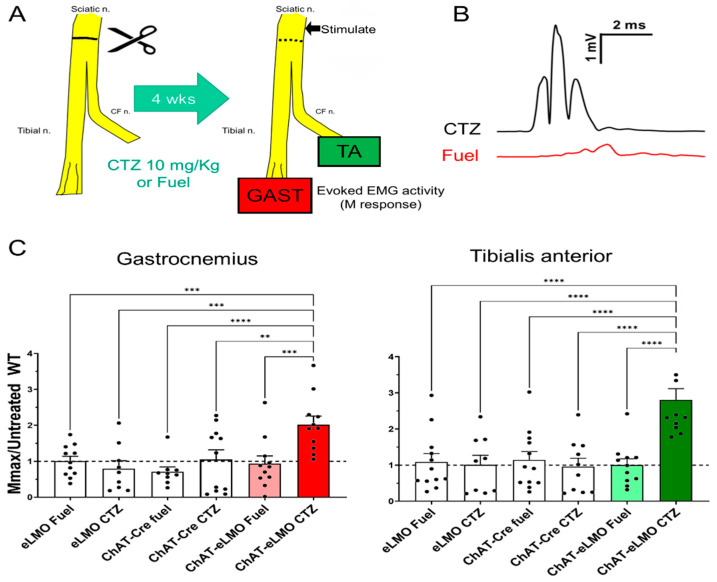
M response recovery in BL-OG treated transgenic mice. (**A**) Mice of each of three genotypes (eLMO3, ChAT-Cre, ChAT-eLMO) were treated once with either coelenterazine (CTZ) or solvent (Fuel) immediately following bilateral transection and repair of the sciatic nerve. The maximal direct muscle (M) response amplitude in the reinnervated gastrocnemius and tibialis anterior muscles evoked by sciatic nerve stimulation was measured four weeks (4 wks) later. (**B**) Examples of maximal M responses (Mmax) recorded from gastrocnemius four weeks after sciatic nerve transection and repair in ChAT-eLMO mice treated either with CTZ or Fuel. (**C**) Mean (±SEM) Mmax responses four weeks after nerve injury are shown for each of the genotypes and treatments studied. Mmax amplitude from each treated mouse was scaled to the average maximal response of a group of untreated WT mice (horizontal dashed line at 1). ** =*p* < 0.01, *** = *p* < 0.0005, **** = *p* < 0.0001.

**Figure 6 ijms-22-07217-f006:**
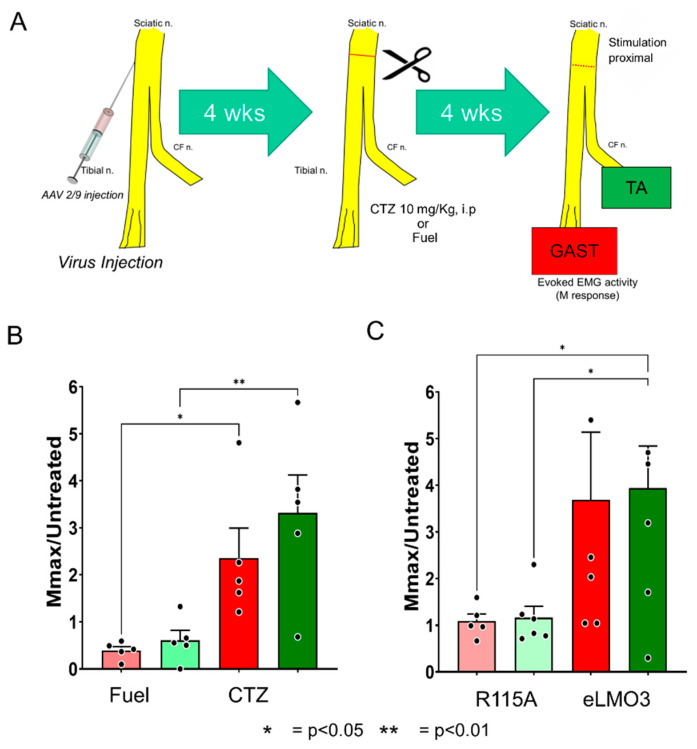
Enhancement of functional motor recovery using virally induced luminopsin. (**A**) Graphical depiction of the experimental protocol. Mouse sciatic nerves were injected with a viral vector encoding an excitatory luminopsin. Four weeks (4 wks) later, the same nerve was cut and surgically repaired. The mouse was then treated with a single dose of CTZ or Fuel. Four weeks later, the recovery of motor innervation was assayed using EMG activity evoked in the gastrocnemius (GAST) and tibialis anterior (TA) muscles produced by electrical stimulation of the sciatic nerve proximal to the injury. (**B**) Sciatic nerves in SLICK-A mice, expressing Cre recombinase in a subset of motoneurons were injected with a viral vector encoding a Cre-dependent luminopsin. The amplitude of maximum evoked EMG activity (Mmax) recorded from the lateral gastrocnemius muscle was scaled to that observed in untreated mice. Mean (±SEM) amplitudes are shown. Lighter bars are from Fuel injected (control) mice. Darker bars are from CTZ injected mice. (**C**) Nerves of WT mice were injected with viral vectors encoding a fully functional luminopsin (eLMO3) (darker bars) or a mutated version in which the channelrhodopsin element was rendered non-functional (R115A) (lighter bars). Mean maximum M responses (±SEM) were recorded 4 weeks after injury and scaled to similar measurements made from untreated control mice. * = *p* < 0.05, ** = *p* < 0.01.

**Figure 7 ijms-22-07217-f007:**
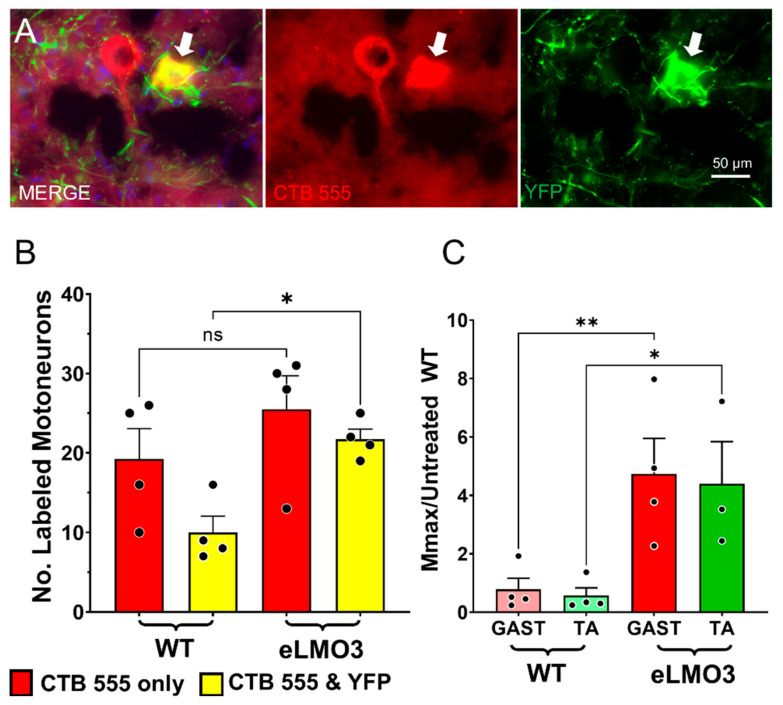
BL-OG treatments increase the participation of motoneurons in axon regeneration following peripheral nerve injury. Sciatic nerves of SLICK-A mice were injected unilaterally with a viral vector encoding a Cre-dependent excitatory luminopsin. Four weeks later, sciatic nerves were cut and repaired bilaterally, and animals treated once with CTZ. Four weeks after that, the extent of successful motor axon regeneration was evaluated using retrograde labeling of motoneurons. (**A**) Two motoneurons marked by a red fluorescent retrograde label (CTB 555) are shown. One of these two neurons (arrows) also expressed yellow fluorescent protein (YFP), indicating that it also expressed Cre-recombinase. (**B**) Mean (±SEM) counts of single-labeled (CTB 555 only) and double-labeled (CTB 555 & YFP) motoneurons are shown on virus injected (eLMO3) and uninjected (WT) sides of the mice. (**C**) Mean (±SEM) Mmax amplitudes recorded from these same mice 4 weeks after nerve injury. Responses on the side of the animals in which eLMO3 was expressed are compared to those in the contralateral (WT) muscles. * = *p* < 0.05, ** = *p* < 0.01.

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
