# Peer review of "Bioluminescent Optogenetics: A Novel Experimental Therapy to Promote Axon Regeneration after Peripheral Nerve Injury"

_ijms, 2021, doi:10.3390/ijms22137217_

Round 1
Reviewer 1 Report
The manuscript describes the results of a study of a new approach to the treatment of axonal injury. In my opinion, the manuscript is very well written. Pictures and text are completely consistent with each other. The results are clearly presented and discussed in the relevant section of the manuscript.
Figure 3 raised one question for me. In Figure 3c, the difference between the bioluminescence level in males and females is quite noticeable. Was this difference statistically estimated? Authors need to discuss this even if the difference is not significant, because the trend is still noticeable.
Author Response
Thank you for this review. We have modified the text on p. 4, lines 136-139 to try to make this finding clearer. The text now reads: "Although bioluminescence amplitude was slightly greater in males than in females, based on the results of our two-way ANOVA, we found no significant sex difference and also found that no significant interaction between sex and amplitude was present (p>0.05; two-way ANOVA; F(1, 12)=4.74, F (2, 12)=0.27, respectively; n=3 each for males and females).
Reviewer 2 Report
The authors present a murine model of nerve injury in which bioluminescent optogenetics (BL-OG) technique based on luminopsins is used to increase neuronal excitability and to promote nerve rigeneration. This topic is really relevant, considering the impact of nerve damage
I have only two question:
The first is related to the animal model of nerve injury: "the sciatic nerve was carefully transected and the freshly cut ends were aligned", so no gap or nerve loss. From a clinical point of view, this represents the most favorable nerve injury, in terms of outcome after surgery. Do the authors think their model could be applicable also in case of a long gap?
The second one is related to the "striking sex difference in the effects
of exercise on axon regeneration following peripheral nerve injury"; there is a possible explanation, that could be clinically relevant, from a translational point of view?
Author Response
Thank you for this review.
To address the first question, we have added text to the Discussion section (p. 11, lines 382-388) to suggest that future studies should examine promoting axon growth across gaps.
With respect to the second question, we have removed the first sentence, quoted by the reviewer, from p. 10, line 13 of the Results section. We think that this only created confusion. We have added a short paragraph to the Discussion section (p. 10, lines 352-356) to indicate the importance of the lack of sex differences in the effects of BL-OG treatment on enhancing axon regeneration in any consideration of clinical applicability of our approach.
Round 2
Reviewer 1 Report
The authors edited the manuscript according to my recommendations.